# Clinical Implications of Serum Hepatitis B Virus Pregenomic RNA Kinetics in Chronic Hepatitis B Patients Receiving Antiviral Treatment and Those Achieving HBsAg Loss

**DOI:** 10.3390/microorganisms9061146

**Published:** 2021-05-26

**Authors:** I-Chin Wu, Wen-Chun Liu, Yen-Cheng Chiu, Hung-Chih Chiu, Pin-Nan Cheng, Ting-Tsung Chang

**Affiliations:** Department of Internal Medicine, National Cheng Kung University Hospital, College of Medicine, National Cheng Kung University, Tainan 70403, Taiwan; wichin@mail.ncku.edu.tw (I.-C.W.); graceliu8911@gmail.com (W.-C.L.); tannoy63352@gmail.com (Y.-C.C.); toad.chiu@gmail.com (H.-C.C.); pncheng@mail.ncku.edu.tw (P.-N.C.)

**Keywords:** entecavir, hepatitis B surface antigen, hepatitis B virus, nucleos(t)ide analogue, pregenomic RNA, viral kinetics

## Abstract

Serum hepatitis B virus (HBV) pregenomic RNA (pgRNA) is correlated with covalently closed circular DNA. We aimed to investigate the utility of serum HBV pgRNA in chronic hepatitis B patients receiving nucleos(t)ide analogue treatment and those achieving HBsAg loss. One hundred and eighty-five patients were enrolled for studying long-term HBV pgRNA kinetics during treatment. Twenty patients achieving HBsAg loss after treatment were enrolled for examining HBV pgRNA kinetics around HBsAg loss. HBV pgRNA significantly decreased in the high baseline HBV pgRNA (≥6 log copies/mL) group but significantly increased in the low baseline HBV pgRNA (<4 log copies/mL) group after 3-month entecavir treatment. Among the 20 patients achieving HBsAg loss, 13 (65%) patients had serum HBV pgRNA higher than the limit of detection (LOD, 1466 copies/mL) when they achieved HBsAg loss. Finally, all 20 patients had HBV pgRNA going below the LOD within 3 years after achieving HBsAg loss. In conclusion, baseline serum HBV pgRNA alone is insufficient for predicting the trajectory of HBV pgRNA. Most patients still had HBV pgRNA higher than the LOD when they achieved HBsAg loss. Further studies on HBV pgRNA kinetics around HBsAg loss would provide an enhanced basis for further applications of HBV pgRNA.

## 1. Introduction

Chronic hepatitis B virus (HBV) infection is an important health problem worldwide. The WHO estimated that 257 million people were living with chronic HBV infection in 2015 [1]. The 5-year cumulative incidence of cirrhosis is 8–20% in untreated patients, and the 5-year cumulative risk of hepatic decompensation is 20% among those with cirrhosis [2]. The annual risk of hepatocellular carcinoma (HCC) is 2–5% in cirrhotic patients [3]. Current treatment guidelines recommend pegylated interferon, entecavir (a nucleoside analog), and tenofovir (a nucleotide analogue) as preferred treatments for chronic HBV infection [2,4,5]. Pegylated interferon treatment achieves hepatitis B e antigen (HBeAg) seroconversion in approximately one-third of patients with HBeAg-positive chronic hepatitis B (CHB), but this treatment is associated with a number of side effects that limit its use [6]. Entecavir and tenofovir are broadly used because of their efficacy, safety, and convenience. However, long-term treatment is necessary to maintain efficacy due to the persistence of covalently closed circular DNA (cccDNA) [7].

Serum HBsAg, which is derived from cccDNA and integrated HBV DNA sequences, could be found in complete virions, subviral particles, empty virions, and HBV RNA-containing virion-like particles [8]. Quantification of serum HBsAg is helpful in defining the phase of HBV infection, identifying the patients most likely to respond to interferon therapy, and determining the likelihood of HBV reactivation after withdrawal of nucleos(t)ide analogue treatment [9]. Hepatitis B core-related antigen (HBcrAg) is a serological marker that simultaneously measures three proteins coded by the precore/core region, including HBeAg, hepatitis B core antigen, and a 22 kDa precore protein (p22cr) [10]. HBcrAg was shown to correlate with intrahepatic cccDNA [11,12] and associated with disease progression [13], HCC development [14,15], treatment responses [16,17], and virological relapse after cessation of treatment [18,19].

HBV pregenomic RNA (pgRNA), which is transcribed from cccDNA, is the template for both reverse transcription of relaxed circular DNA and translation of viral polymerase and core proteins [20]. Circulating HBV pgRNA could be found in the blood of CHB patients. Previous studies have shown that serum HBV pgRNA is encapsidated and is present in virion-like particles [21,22]. HBV pgRNA might have diagnostic advantages because it carries viral genetic information and because its quantification is not affected by immune complexes of antibodies and viral antigens [23]. Several studies have shown that serum HBV pgRNA is correlated with intrahepatic cccDNA [23,24,25] and is associated with viral rebound after withdrawal of treatment [22,26,27,28]. However, the serum HBV pgRNA kinetics in CHB patients receiving long-term nucleos(t)ide analogue therapy and those achieving HBsAg loss are not fully understood. This study investigated (1) long-term HBV pgRNA kinetics during entecavir treatment, (2) the factors associated with virological relapse after cessation of entecavir therapy, (3) serum HBV pgRNA kinetics in the patients achieving HBsAg loss, and (4) the factors associated with HBsAg loss.

## 2. Materials and Methods

### 2.1. Patient Enrollment

Patients were retrospectively enrolled from our clinics, previous clinical trials, and databases in National Cheng Kung University Hospital. All the CHB patients in Part I and Part II received at least 2 years of consecutive entecavir treatment, which was the only antiviral therapy during the study period. The indication for antiviral therapy mostly followed the Asian Pacific Association for the Study of the Liver HBV treatment guideline [5]. The exclusion criteria were as follows: (1) treatment history with nucleos(t)ide analogues or interferon, (2) coinfection with hepatitis C virus or human immunodeficiency virus, (3) end-stage renal disease, (4) systemic chemotherapy or immunotherapy due to malignancy, and (5) post-organ transplantation [29]. Finally, 185 CHB patients were enrolled for studying the long-term kinetics of serum HBV pgRNA during entecavir therapy (Part I) and 47 CHB patients for analyzing the factors associated with virological relapse after cessation of entecavir therapy (Part II), 55 CHB patients who had achieved HBsAg loss, 20 healthy subjects with negative HBsAg but positive antibody to hepatitis B core antigen (anti-HBc), and 17 healthy controls with negative HBsAg and negative anti-HBc for determining the expression of serum HBV pgRNA after HBsAg loss (Part III). 20 CHB patients received nucleos(t)ide analogue treatment with HBsAg loss for investigating HBV pgRNA kinetics in the patients achieving HBsAg loss (Part IV), 20 CHB patients received nucleos(t)ide analogue treatment with HBsAg loss, and 141 CHB patients received nucleos(t)ide analogue treatment without HBsAg loss for assessing the factors associated with HBsAg loss (Part V) (Figure 1). This study was approved by the Institutional Review Board of National Cheng Kung University Hospital. Informed consents were provided by study participants.

### 2.2. Patient Monitoring

The patients in Part I and II were followed up at 3- to 6-month (12- to 24-week) intervals through liver biochemistry and serum HBV DNA tests, as well as abdominal sonography. In HBeAg-positive patients, HBeAg was assessed every 3 to 6 months until negative results were obtained. Liver cirrhosis was diagnosed by liver biopsy, abdominal sonography, computed tomography (CT), magnetic resonance imaging (MRI), or portal hypertension (esophageal or cardiac varices revealed by esophagogastroduodenoscopy). Hepatocellular carcinoma was diagnosed by histological examination or dynamic imaging studies (CT or MRI). The upper limit of normal (ULN) of alanine aminotransferase (ALT) was 50 U/mL in male patients and 35 U/mL in female patients at National Cheng Kung University Hospital.

For studying the kinetics of serum HBV pgRNA, HBV DNA, and HBsAg during entecavir therapy, serum HBV pgRNA and HBV DNA quantification was performed at baseline and the 3rd month (12th week), 6th month (24th week), 12th month (48th week), and 60th month (240th week) of treatment, and serum HBsAg quantification was performed at baseline and the 1st year (48th week) and 5th year (240th week) of treatment. Virological response was defined as undetectable serum HBV DNA (<60 IU/mL) after entecavir therapy.

For analyzing the factors associated with virological relapse, these patients were monitored for up to 3 years after cessation of entecavir therapy. Serum HBV pgRNA, HBV DNA, and HBsAg levels were determined at the end of treatment. Virological relapse was defined as serum HBV DNA level >2000 IU/mL after cessation of entecavir therapy.

### 2.3. Quantification of Serum HBsAg

Serum HBsAg levels were measured using the Architect HBsAg assay (Abbott, Chicago, IL, USA), with a limit of detection (LOD) of 0.05 IU/mL. For statistical analysis, serum samples with HBsAg levels below LOD were recorded as 0.04 IU/mL (−1.40 log IU/mL).

### 2.4. Quantification of Serum HBV DNA

Serum HBV DNA levels were determined using the Roche Cobas Amplicor (LOD: 60 IU/mL); Roche Cobas TaqMan 48 analyzer (LOD: 29 IU/mL); Roche Cobas AmpliPrep/Cobas TaqMan HBV Test, version 1.0 (LOD: 12 IU/mL); and Roche Cobas AmpliPrep/Cobas TaqMan HBV Test, version 2.0 (LOD: 20 IU/mL). For statistical analysis, serum samples with HBV DNA values below LOD and serum samples in which HBV DNA was not detected were recorded as LOD – 1 IU/mL and 1 IU/mL, respectively.

### 2.5. Extraction and Reverse Transcription of HBV pgRNA

HBV RNA was extracted from 150 μL of serum using the Total RNA Extraction Miniprep System Kit according to the manufacturer’s instructions (Viogene, Taipei, Taiwan) and was treated with DNase I (Thermo Fisher Scientific, Waltham, MA, USA). Isolated HBV RNA was reverse transcribed using RevertAid reverse transcriptase (Thermo Fisher Scientific, Waltham, MA, USA) with an HBV pgRNA-specific reverse transcription (RT) primer (Appendix A). Before commencing RT, 20 μL of RNA, 1 μL of 100 μM primer, and 1 μL of 10 mM dNTPs (Thermo Fisher Scientific, Waltham, MA, USA) were mixed, incubated at 70 °C for 5 min, and then placed immediately on ice for 1 min. RT was then initiated with the addition of the RT reaction mixture to ensure a final volume of 35 μL at a final concentration of 1× RT buffer, 1 μL of RNAse inhibitor (Life Technologies), and 1 μL of RevertAid reverse transcriptase (Life Technologies). Cycling conditions were 42 °C for 60 min, followed by 75 °C for 5 min. cDNA samples were maintained at 4 °C before proceeding to quantitative real-time polymerase chain reaction (qPCR).

### 2.6. Quantification of Serum HBV pgRNA

Serum HBV pgRNA levels were detected by qPCR on the LightCycler 480 II Real-Time PCR Detection System (Roche, Mannheim, Germany) by using a SYBR Green method. The primers used to detect 3.5 kb HBV pgRNA are provided in Appendix A. PCR was used to construct the standards by using each primer from the HBV full genome (accession number: KJ790199) [30], and the PCR products were subsequently ligated into the T&A™ Cloning Vector (Yeastern Biotech, New Taipei, Taiwan). The qPCR reaction mixture (20 μL) contained 10 μL of 2× GoTaq^®^ Green Master Mix (Promega Corp., Madison, WI, USA), 0.5 μL of the forward primer (10 μM), 0.5 μL of the reverse primer (10 μM), 1 μL of the cDNA template, and 8 μL of double distilled water (ddH_2_O). The reaction mixture was denatured at 95 °C for 5 min, followed by 40 cycles at 95 °C for 20 s and 60 °C for 40 s. The LOD of serum HBV pgRNA was 1466 copies/mL, as calculated by probit analysis (Appendix A). For statistical analysis, 1465 copies/mL (3.17 log copies/mL) was recorded as the value for serum samples with HBV pgRNA below LOD or not detected.

### 2.7. HBV Genotyping

The HBV genotype was determined through melting curve analysis using LightCycler hybridization probes, as described previously [31].

### 2.8. Statistical Analysis

Continuous variables were compared between two groups using Student’s *t*-test or the Mann–Whitney test, as appropriate. Continuous variables were compared between three groups using one-way ANOVA or the Kruskal–Wallis test, as appropriate. The distributions of categorical variables were compared using Pearson’s chi-squared test or Fisher’s exact test when the expected value was less than 5 in 2 × 2 tables. The cumulative incidence of virological response, virological relapse, and HBsAg loss was derived using the Kaplan-Meier analysis and was tested using the log-rank test. Multivariate analysis was performed using Cox proportional hazards regression to identify the factors associated with virological responses, virological relapse, and HBsAg loss. Linear mixed models with a random intercept were used for analyzing the longitudinal changes in serum HBV pgRNA, HBsAg, and HBV DNA levels. In this model, groups and time points were considered categorical variables and represented by dummy variables. Statistical analysis was performed using Stata 16.1 (StataCorp, College Station, TX, USA). The results were considered statistically significant at *p* < 0.05.

## 3. Results

### 3.1. Clinical Characteristics of Patients in the Analysis of HBV pgRNA Kinetics during Entecavir Therapy

In Part I, 185 CHB patients were enrolled for the analysis of long-term HBV pgRNA kinetics during entecavir therapy. The mean age was 51.0 ± 12.0 years, 70.3% were men, and 29.2% were HBeAg-positive. At baseline, the mean ALT, HBsAg, HBV DNA, and HBV pgRNA levels were 3.84 ± 5.84 × ULN, 3.15 ± 0.78 log IU/mL, 5.83 ± 1.72 log IU/mL, and 5.07 ± 1.98 log copies/mL, respectively. For patients with available data on HBV genotypes, 88 had genotype B and 76 had genotype C. The median treatment duration was 4.85 (range, 1.84–11.29) years. As expected, HBeAg-positive patients were younger and had higher baseline HBsAg, higher baseline HBV DNA, higher proportion of genotype C, lower proportion of liver cirrhosis, and later virological responses than HBeAg-negative patients (Table 1).

### 3.2. HBV pgRNA Kinetics during Entecavir Treatment

For analyzing serum HBV pgRNA kinetics during entecavir treatment, patients were categorized into three groups according to baseline HBV pgRNA levels: high, medium, and low baseline HBV pgRNA groups (baseline HBV pgRNA levels of ≥6, ≥4 and <6, and <4 log copies/mL, respectively). In the high baseline HBV pgRNA group, serum HBV pgRNA levels decreased rapidly in the first 3 months and remained constant thereafter. In the medium baseline HBV pgRNA group, HBV pgRNA levels decreased gradually in the first 12 months and remained constant thereafter. In the low baseline HBV pgRNA group, HBV pgRNA levels increased in the first 3 months and remained constant thereafter. Compared with the medium and low HBV pgRNA groups, the high baseline HBV pgRNA group had higher HBV pgRNA levels at baseline, 12 months, and 60 months. The high baseline HBV pgRNA group also had higher HBV pgRNA levels at 3 months and 6 months than the low HBV pgRNA group. The medium baseline HBV pgRNA group had higher HBV pgRNA levels at baseline and 3 months compared with the low HBV pgRNA group. No significant differences were detected in HBV pgRNA levels at 6 months, 12 months, and 60 months between the medium and low HBV pgRNA groups (Figure 2).

Figure 3 shows the serum HBV DNA kinetics during entecavir treatment, categorized by baseline serum HBV pgRNA levels. HBV DNA levels decreased rapidly in the first 3 months and decreased gradually thereafter. The high baseline HBV pgRNA group had higher HBV DNA levels at baseline than did the medium HBV pgRNA group. No significant differences were observed in HBV DNA levels at 3 months, 6 months, 12 months, or 60 months between the high, medium, and low HBV pgRNA groups.

Figure 4 presents the serum HBsAg kinetics during entecavir treatment, categorized by baseline serum HBV pgRNA levels. In the high baseline HBV pgRNA group, HBsAg levels decreased rapidly in the first 12 months and remained constant thereafter. In the medium and low baseline HBV pgRNA groups, HBsAg levels decreased gradually after entecavir therapy initiation. The high baseline HBV pgRNA group had higher HBsAg at baseline compared with the medium and low HBV pgRNA groups. No significant differences were observed in HBsAg levels at 12 months and 60 months between the high, medium, and low HBV pgRNA groups.

Moreover, no significant differences were found in age, sex, HBeAg status, HBV genotype, baseline ALT, and liver cirrhosis status between the high, medium, and low HBV pgRNA groups (Appendix A).

### 3.3. Baseline Serum HBV pgRNA and Virological Response

The virological response to entecavir treatment was assessed in 171 CHB patients, after excluding 8 HBeAg-positive and 6 HBeAg-negative patients with limited serum samples within the first year, as the limited serum samples prevented the virological response from being precisely determined. Kaplan-Meier analysis showed that baseline HBV pgRNA < 6.4 log copies/mL was associated with earlier virological responses compared with baseline HBV pgRNA ≥ 6.4 log copies/mL (*p* = 0.0018, Figure 5). Multivariate analysis showed that younger age, HBeAg negativity, and lower baseline HBV pgRNA were independently associated with earlier virological responses (age < 50 years vs. ≥ 50 years old: hazard ratio (HR): 1.54, 95% confidence interval (CI): 1.03–2.30, *p* = 0.034; HBeAg negativity vs. positivity: HR: 2.03, 95% CI: 1.27–3.25, *p* = 0.003; HBV pgRNA < 6.4 vs. ≥ 6.4 log copies/mL: HR: 1.57, 95% CI: 1.06–2.31, *p* = 0.023, Table 2).

### 3.4. Correlation between Serum HBV pgRNA, HBV DNA, and HBsAg

Before entecavir treatment, HBV pgRNA was positively correlated with HBV DNA and HBsAg in HBeAg-positive patients (*r* = 0.28, *p* = 0.04 and *r* = 0.31, *p* = 0.025, respectively). By contrast, HBV pgRNA was not correlated with HBV DNA or HBsAg in HBeAg-negative patients. HBV DNA was positively correlated with HBsAg in both HBeAg-positive and HBeAg-negative patients (*r* = 0.58, *p* < 0.00001 and *r* = 0.47, *p* < 0.00001, respectively). After 1 year of entecavir treatment, HBV pgRNA was not correlated with HBsAg in either HBeAg-positive or HBeAg-negative patients (Appendix A).

### 3.5. Factors Associated with Virological Relapse after Cessation of Entecavir Therapy

In Part II, 47 patients with CHB were enrolled for analyzing the factors associated with virological relapse after cessation of entecavir therapy. Of these patients, 74.5% were men and 55.3% were HBeAg-positive before receiving entecavir treatment. The median treatment duration was 4.32 (range, 2.00–8.22) years. At the end of treatment, the mean age, ALT, HBsAg, and HBV pgRNA levels were 46.8 ± 11.9 years, 0.74 ± 0.85 × ULN, 2.76 ± 1.30 log IU/mL, and 3.90 ± 0.97 log copies/mL, respectively. The median HBV DNA level was 0.00 (range, 0.00–1.84) log IU/mL at the end of treatment. For the patients with available data on HBV genotypes, 20 had genotype B and 24 had genotype C. Compared with HBeAg-negative patients, HBeAg-positive patients were younger and had higher baseline HBsAg, baseline HBV DNA, and HBsAg levels at the end of treatment (Appendix A).

Kaplan-Meier analysis showed that “HBV pgRNA ≥ 1466 copies/mL” and “HBsAg ≥ 2 log IU/mL or HBV pgRNA ≥ 1466 copies/mL” at the end of treatment were associated with earlier virological relapse after cessation of entecavir therapy (*p* = 0.040 and *p* = 0.038, Appendix A). Multivariate analysis showed that “HBsAg ≥ 2 log IU/mL or HBV pgRNA ≥ 1466 copies/mL” at the end of treatment was independently associated with earlier virological relapse after cessation of entecavir therapy (HR: 4.26, 95% CI: 1.08–16.89, *p* = 0.039, Appendix A).

### 3.6. Expression of Serum HBV pgRNA after HBsAg Loss

In Part III, 34 CHB patients who achieved HBsAg loss after nucleos(t)ide analogue treatment, 21 CHB patients who achieved HBsAg loss spontaneously, 20 healthy subjects with negative HBsAg but positive anti-HBc, and 17 healthy controls with negative HBsAg and negative anti-HBc were enrolled for determining the expression of serum HBV pgRNA after HBsAg loss. Serum HBV pgRNA was not detected in CHB patients who achieved HBsAg loss spontaneously, healthy subjects with negative HBsAg but positive anti-HBc, or healthy controls with negative HBsAg and negative anti-HBc. Notably, 10 CHB patients who received nucleos(t)ide analogue had serum HBV pgRNA higher than the LOD (1466 copies/mL) at or after HBsAg loss, with a median HBV pgRNA of 5.80 (range, 4.81–7.34) log copies/mL (Appendix A).

### 3.7. Serum HBV pgRNA Kinetics in CHB Patients Who Achieved HBsAg Loss after Nucleos(t)ide Analogue Treatment

In Part IV, 20 CHB patients who achieved HBsAg loss after nucleos(t)ide analogue treatment with serial serum samples before and after HBsAg loss were enrolled. Five (25%) and two (10%) patients had HBV pgRNA going below the LOD (1466 copies/mL) before and at HBsAg loss, respectively. The other 13 (65%) patients had serum HBV pgRNA higher than the LOD when they achieved HBsAg loss, with a median HBV pgRNA of 5.90 (range, 4.81–7.34) log copies/mL. Finally, all of these 20 (100%) patients had HBV pgRNA going below the LOD within 3 years after achieving HBsAg loss (Figure 6 and Table 3). Anti-HBs became positive during HBsAg loss in one (5%) patient and after HBsAg loss in 19 (95%) patients (Figure 6). Longitudinal changes in HBsAg, HBV DNA, HBV pgRNA, and antibody to hepatitis B surface antigen (anti-HBs) are illustrated in Appendix A.

### 3.8. Factors Associated with HBsAg Loss after Nucleos(t)ide Analogue Treatment

In Part V, a total of 161 CHB patients were enrolled for analyzing the factors associated with HBsAg loss after nucleos(t)ide analogue treatment (Table 3). Kaplan-Meier analysis revealed that a HBsAg decline of ≥1.5 log IU/mL within the first year was associated with HBsAg loss (*p* < 0.0001, Figure 7). Multivariate analysis revealed that younger age and greater HBsAg decline within the first year were independently associated with HBsAg loss (age < 50 years vs. ≥ 50 years old: HR: 9.81, 95% CI: 1.23–77.94, *p* = 0.031; HBsAg decline ≥1.5 log IU/mL vs. HBsAg decline <1.5 log IU/mL: HR: 53.59, 95% CI: 3.83–749.5, *p* = 0.003, Table 4).

## 4. Discussion

The study was the largest one to investigate long-term HBV pgRNA kinetics in CHB patients receiving nucleos(t)ide analogue treatment and the first one to study HBV pgRNA kinetics in those patients achieving HBsAg loss.

Serum HBV pgRNA levels decreased rapidly in the first 3 months in the high baseline HBV pgRNA group. The levels decreased gradually in the first 12 months in the medium baseline HBV pgRNA group. By contrast, the levels increased in the first 3 months in the low baseline HBV pgRNA group. The high baseline HBV pgRNA group had higher HBV pgRNA levels at baseline, 12 months, and 60 months than the medium and low HBV pgRNA groups. No significant differences were observed in HBV pgRNA levels at 6 months, 12 months, and 60 months between the medium and low HBV pgRNA groups. Our study demonstrated that baseline serum HBV pgRNA alone is insufficient for predicting the trajectory of HBV pgRNA. A more accurate description of HBV pgRNA trajectory would be provided by serial HBV pgRNA levels at baseline, 6 months, and 12 months after treatment.

HBV pgRNA is the template for reverse transcription and is normally degraded during the process of reverse transcription. Nucleos(t)ide analogues inhibit reverse transcription. While reverse transcription is inhibited, HBV pgRNA accumulates and is released in the circulation as HBV pgRNA-containing viral particles. Wang et al. showed the levels of HBV pgRNA virion increased after entecavir treatment in cell culture models and HBV transgenic mice [22]. Goncalves et al. developed a mathematical model that predicted a transient increase of HBV RNA levers during the first week of tenofovir treatment followed by a slower decline [32]. Our study illustrated that HBV pgRNA significantly increased within 3 months of entecavir treatment and remained higher than the baseline level in the low baseline HBV pgRNA group. We suppose longer antiviral treatment is needed to induce HBV pgRNA lower than the baseline in this group, especially since a significant proportion of them achieved HBsAg loss. Larger and more detailed kinetic studies are needed to characterize HBV pgRNA kinetics under nucleos(t)ide analogue treatment, which would help us to understand HBV–host interactions and nucleos(t)ide analogues’ mode of actions [33].

Even more remarkably, among the 20 patients achieving HBsAg loss, 13 (65%) patients had serum HBV pgRNA higher than the LOD (1466 copies/mL) when they achieved HBsAg loss, with a median HBV pgRNA of 5.90 (range, 4.81–7.34) log copies/mL. Finally, all of these 20 patients had HBV pgRNA going below the LOD within 3 years after achieving HBsAg loss. Mak et al. showed that 3 of 19 (15.8%) treatment-naive patients with HBsAg loss had detectable serum HBV RNA and HBcrAg [34]. These findings suggest that HBsAg loss does not necessarily equal complete cessation of viral transcription and translation. The sensitivity of quantitative assays, HBsAg escape mutants, and interference of HBsAg detection through immune complexes with anti-HBs in excess might be the reasons why cccDNA is still transcriptionally active in patients with undetectable HBsAg [35]. Further studies regarding serum HBV pgRNA kinetics in CHB patients achieving HBsAg loss and the time when HBV pgRNA becomes undetectable would improve the role of HBV pgRNA in clinical practice and its use as a surrogate marker of cccDNA.

Previous studies have shown that serum HBV pgRNA was positively correlated with HBV DNA before antiviral treatment in both HBeAg-positive and HBeAg-negative patients [21,24,36]. One study demonstrated that the correlation between HBV pgRNA and HBsAg before treatment was moderate in HBeAg-positive patients but weak in HBeAg-negative patients [36]. Another study showed that serum HBV pgRNA was positively correlated with HBsAg before treatment in HBeAg-positive patients, but not in HBeAg-negative patients [24]. Two studies have reported that serum HBV pgRNA was positively correlated with HBsAg after nucleoside analogue therapy [23,37]. Our study revealed that HBV pgRNA was positively correlated with HBV DNA and HBsAg before treatment in HBeAg-positive patients, but not in HBeAg-negative patients. The discrepancy may be explained by differences in patient characteristics, HBV genotypes, the presence of the HBV basal core promoter mutation, impaired virion productivity in HBeAg-negative patients, the proportion of HBsAg derived from the integrated HBV genome, and detection methods [20,21,36,38,39].

Several studies have shown that serum HBV pgRNA was associated with treatment responses in patients with CHB receiving pegylated interferon or nucleos(t)ide analogue therapy [40,41,42,43]. Some studies have demonstrated that serum HBV pgRNA was associated with virological relapse after discontinuation of nucleos(t)ide analogue therapy [22,26,27,28]. Seto et al. showed that serum HBV pgRNA is useful for deciding on entecavir cessation in patients with CHB, especially with low HBsAg levels [44]. Our study demonstrated that younger age, HBeAg negativity, and lower baseline HBV pgRNA were independently associated with earlier virological responses in patients receiving entecavir treatment, and that “HBsAg ≥ 2 log IU/mL or HBV pgRNA ≥ 1466 copies/mL” at the end of treatment was independently associated with earlier virological relapse after cessation of entecavir therapy. All of these findings indicate that serum HBV pgRNA could serve as a biomarker for monitoring the efficacy of antiviral therapy.

Our study revealed that age below 50 years and HBsAg decline greater than 1.5 log IU/mL within the first year were independently associated with HBsAg loss after nucleos(t)ide analogue therapy. These findings are compatible with previous studies. Lee et al. indicated that younger age, lower baseline HBsAg, rapid HBsAg decline at week 24, and achievement of sustained virological response were predictors of HBsAg loss after peginterferon therapy [45]. Jeng et al. demonstrated that shorter time to undetectable HBV DNA, greater HBsAg reduction during therapy, lower end-of-treatment HBsAg level, patients with sustained response, and untreated relapsers were predictors for HBsAg loss after cessation of nucleos(t)ide analogue therapy [46]. One recent study also demonstrated HBsAg to be superior to hepatitis B core-related antigen and HBV pgRNA in predicting HBsAg loss with peginterferon therapy [47].

Bai et al. showed that extracellular HBV RNAs comprised full-length pgRNA and 3′ RNA fragments degraded by the RNase H domain of polymerase from incomplete RT. These RNAs were localized in naked capsids and virions in cell culture supernatants, and they circulated as unenveloped capsids in the form of capsid–antibody complexes and virions in the blood of hepatitis B patients [48]. Another study showed that HBV RNA virion-like particles produced under nucleos(t)ide analogue treatment also comprised full-length pgRNA and short RNA species. These short RNA species consisted of pgRNA splicing variants and 3′-terminal truncations induced by nucleos(t)ide analogue treatment; the truncations might be a major contributor to the replication deficiency and lack of infectivity of HBV RNA virion-like particles produced under nucleos(t)ide analogue treatment [49]. Anderson et al. showed that circulating HBV pgRNA is primarily full-length in patients receiving nucleos(t)ide analogue therapy [50]. Besides, Freitas et al. showed that serum HBV RNA derived not only from cccDNA but also from integrated HBV DNA sequences [51]. Further studies that clarify the production mechanisms of HBV pgRNA-containing viral particles and determine the expression of cccDNA-derived HBV RNA and integrant-derived HBV RNA in CHB patients are needed for further application of HBV RNA as a biomarker.

Our study has a few limitations. First, this was a retrospective study using stored serum samples. Second, only Asian patients infected with HBV genotype B or C were included. Third, serum HBV pgRNA levels were determined using an in-house method, which utilized DNase I for removal of viral DNA, with an LOD of 1466 copies/mL. Future prospective studies using commercial kits with a higher specificity and a lower LOD in large and diverse populations would provide more comprehensive descriptions of HBV pgRNA kinetics during antiviral treatment, especially for patients with low serum HBV pgRNA levels [52,53]. Fourth, HBsAg loss was determined by the Abbott Architect HBsAg assay (cut-off value: 0.05 IU/mL) in this study. Future studies using a more sensitive qualitative assay or an ultrahigh sensitivity HBsAg assay (ICT-CLEIA, cutoff value: 0.0005 IU/mL) would provide a more accurate assessment of functional cure [54,55].

In summary, our study demonstrated the kinetics and clinical utility of serum HBV pgRNA in CHB patients receiving long-term entecavir therapy and those achieving HBsAg loss. Baseline serum HBV pgRNA alone is insufficient for predicting the trajectory of HBV pgRNA. Serum HBV pgRNA was associated with virological responses during treatment and virological relapse after treatment withdrawal. Moreover, most patients had serum HBV pgRNA higher than the LOD (1466 copies/mL) when they achieved HBsAg loss. Further studies clarifying the production mechanisms of viral particles containing HBV pgRNA would provide an enhanced basis for further applying serum HBV pgRNA as a biomarker in antiviral treatment.

## Figures and Tables

**Figure 1 microorganisms-09-01146-f001:**
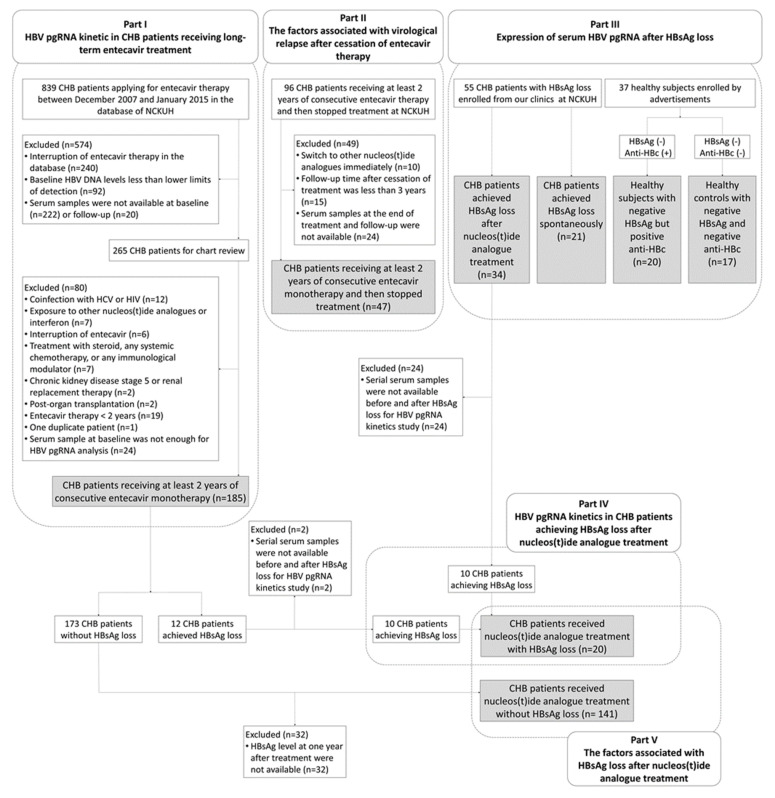
Patient flow diagram. Abbreviations: Anti-HBc, antibody to hepatitis B core antigen; CHB, chronic hepatitis B; HBsAg, hepatitis B surface antigen; HBV, hepatitis B virus; HCV, hepatitis C virus; HIV, human immunodeficiency viruses; NCKU, National Cheng Kung University.

**Figure 2 microorganisms-09-01146-f002:**
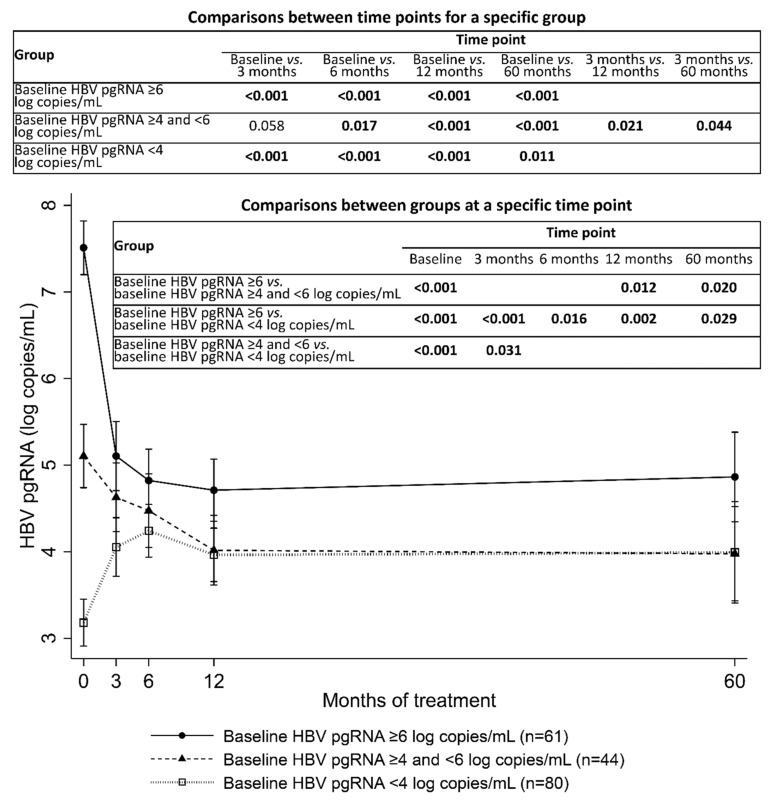
Serum HBV pgRNA kinetics during entecavir treatment, categorized by baseline serum HBV pgRNA levels. Error bars represent 95% confidence intervals. Significant *p* values and one borderline *p* value are presented.

**Figure 3 microorganisms-09-01146-f003:**
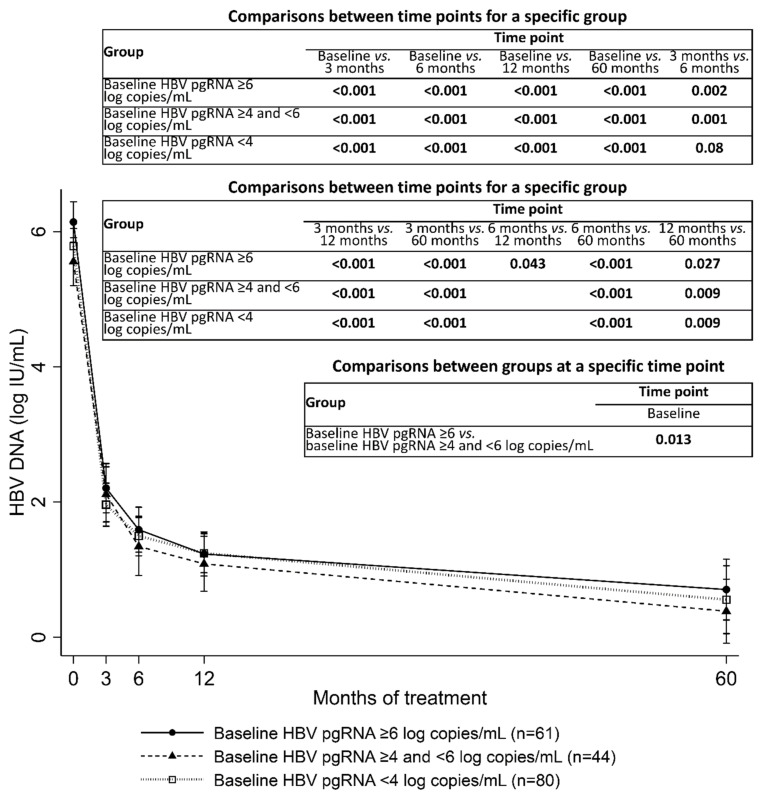
Serum HBV DNA kinetics during entecavir treatment, categorized by baseline serum HBV pgRNA levels. Error bars represent 95% confidence intervals. Significant *p* values are presented.

**Figure 4 microorganisms-09-01146-f004:**
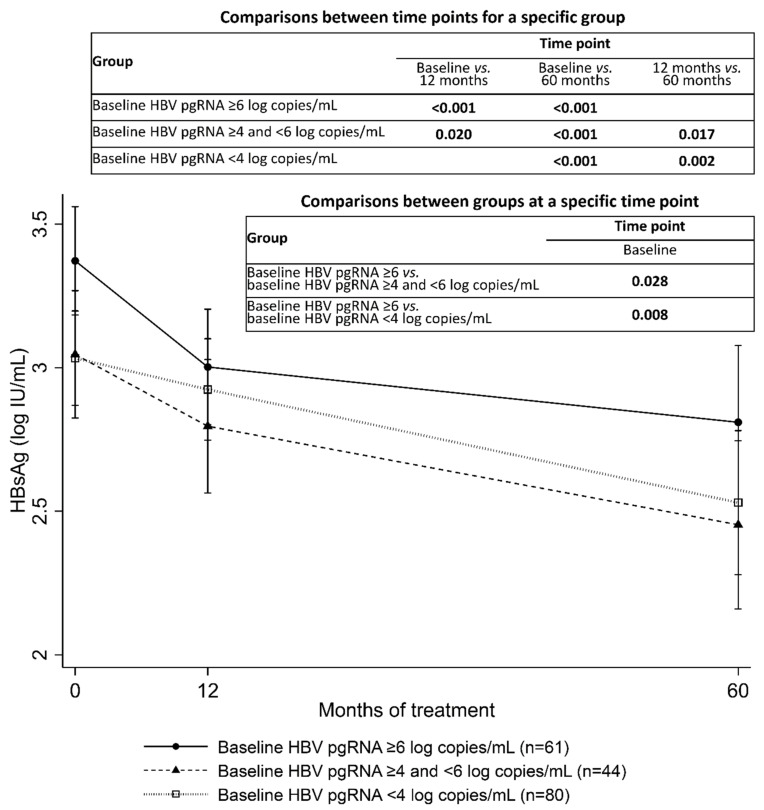
Serum HBsAg kinetics during entecavir treatment, categorized by baseline serum HBV pgRNA levels. Error bars represent 95% confidence intervals. Significant *p* values are presented.

**Figure 5 microorganisms-09-01146-f005:**
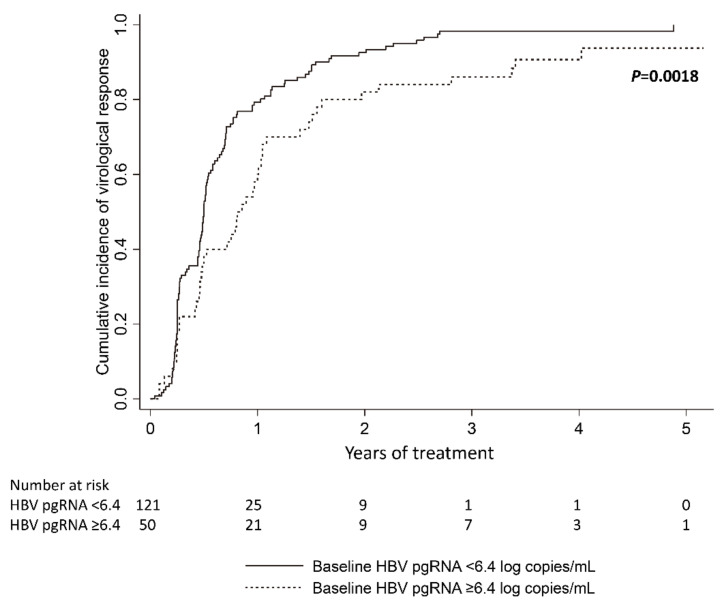
Cumulative incidence of virological response during entecavir treatment.

**Figure 6 microorganisms-09-01146-f006:**
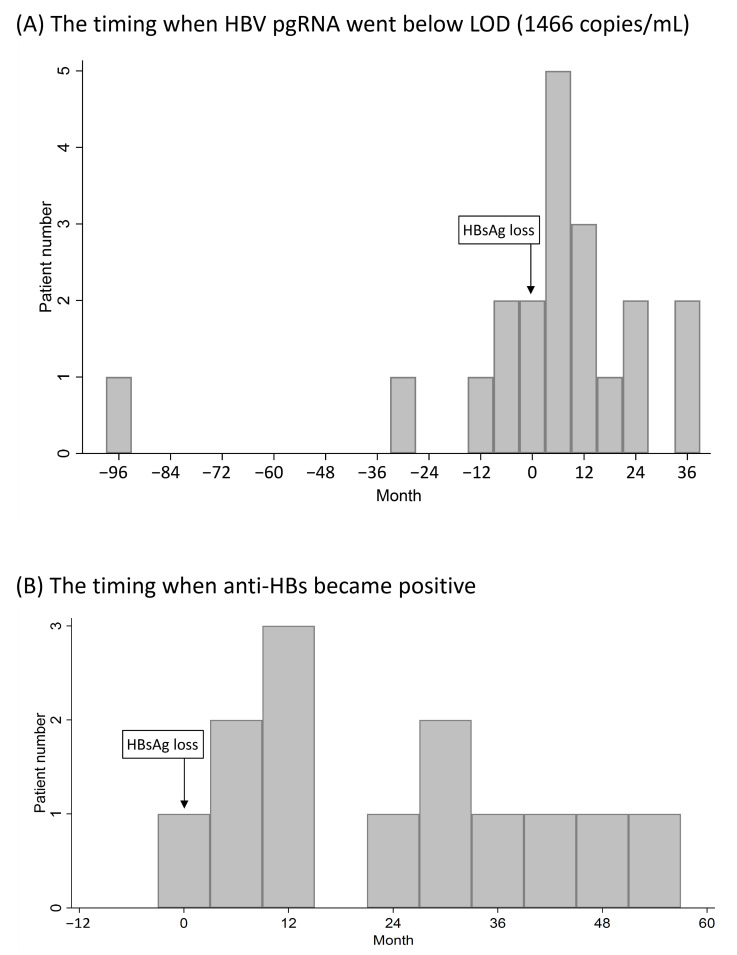
The timings when HBV pgRNA went below LOD (1466 copies/mL) (**A**) and anti-HBs became positive (**B**) in chronic hepatitis B patients who achieved HBsAg loss after nucleos(t)ide analogue treatment.

**Figure 7 microorganisms-09-01146-f007:**
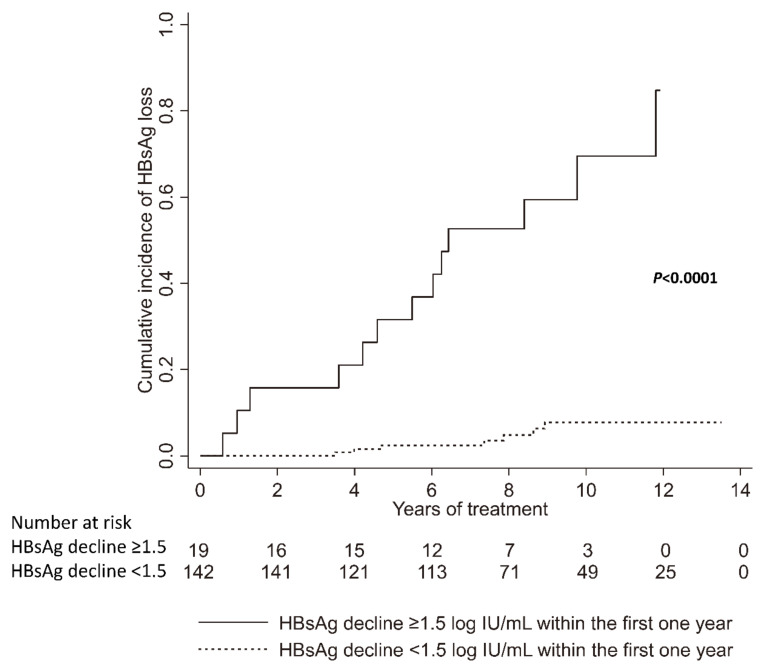
Cumulative incidence of HBsAg loss in chronic hepatitis B patients receiving nucleos(t)ide analogue treatment.

**Table 1 microorganisms-09-01146-t001:** Clinical characteristics of 185 patients with chronic hepatitis B for studying the kinetics of serum HBV pgRNA during entecavir therapy, categorized by HBeAg status.

Characteristics	Total Patients (*n* = 185)	HBeAg-Positive Patients (*n* = 54)	HBeAg-Negative Patients (*n* = 131)	*p* ^a^
Age (year)	51.0 ± 12.0	43.8 ± 12.5	54.0 ± 10.5	**<0.0001**
Male (%)	130/185 (70.3%)	36/54 (66.7%)	94/131 (71.8%)	0.49
HBV genotype (B:C) ^b^	88:76	17:36	71:40	**<0.001**
Baseline ALT (× ULN)	3.84 ± 5.84	4.12± 6.63	3.73 ± 5.51	0.68
Baseline HBsAg (log IU/mL)	3.15 ± 0.78	3.78 ± 0.72	2.89 ± 0.65	**<0.0001**
Baseline HBV DNA (log IU/mL)	5.83 ± 1.72	7.29 ± 1.32	5.23 ± 1.49	**<0.0001**
Baseline HBV pgRNA (log copies/mL)	5.07 ± 1.98	5.48 ± 2.03	4.89 ± 1.94	0.066
Liver cirrhosis (%)	62/185 (33.5%)	12/54 (22.2%)	50/131 (38.2%)	**0.037**
HCC (%) ^c^	29/185 (15.7%)	5/54 (9.3%)	24/131 (18.3%)	0.12
Duration of entecavir therapy (years)	4.85 (1.84–11.29)	4.43 (1.95–8.85)	4.92 (1.84–11.29)	0.93
Virological response rate (%) ^d^	166/171 (97.1%)	42/46 (91.3%)	124/125 (99.2%)	**0.019**
Time to virological response (years)	0.51 (0.04–4.88)	1.01 (0.22–4.02)	0.46 (0.04–4.88)	**0.0001**
HBeAg seroconversion rate (%)		19/54 (35.2%)		
Time to HBeAg seroconversion (years)		1.24 (0.21–7.49)		
Virological response and HBeAg seroconversion rate (%)		15/45 (33.3%)		
Time to virological response and HBeAg seroconversion (years)		1.07 (0.49–4.06)		

Continuous variables are expressed as mean ± standard deviation, except for duration of entecavir therapy, time to virological response, time to HBeAg seroconversion, and time to virological response and HBeAg seroconversion, which are expressed as median (range). Significant *p* values are presented in bold. ^a^
*p* value for HBeAg-positive patients compared with HBeAg-negative patients. ^b^ HBV genotype was not determined in 1 HBeAg-positive patient and 20 HBeAg-negative patients because of low baseline HBV viral loads. ^c^ HCC was diagnosed before or within half a year of entecavir therapy. ^d^ The virological response (serum HBV DNA <60 IU/mL) was not assessed in eight HBeAg-positive patients and six HBeAg-negative patients because limited serum samples within the first year prevented the virological response from being determined precisely.

**Table 2 microorganisms-09-01146-t002:** Univariate and multivariate analyses of factors associated with virological response during entecavir therapy.

Factors	Univariate Analysis			Multivariate Analysis		
	HR	95% CI	*p* Value	HR	95% CI	*p*
Age (<50 vs. ≥50 years old)	0.91	0.67–1.23	0.54	1.54	1.03–2.30	**0.034**
Sex (male vs. female)	0.90	0.65–1.25	0.53	0.73	0.50–1.07	0.11
Baseline HBeAg (negative vs. positive)	2.61	1.81–3.77	**<0.001**	2.03	1.27–3.25	**0.003**
HBV genotype (C vs. B)	0.86	0.62–1.19	0.36	0.96	0.64–1.42	0.83
ALT (×ULN)	0.99	0.96–1.02	0.55	0.99	0.96–1.03	0.64
Baseline HBsAg (<4 vs. ≥4 log IU/mL)	2.56	1.57–4.17	**<0.001**	1.74	0.93–3.25	0.08
Baseline HBV DNA (<5 vs. ≥5 log IU/mL)	1.98	1.44–2.74	**<0.001**	1.28	0.86–1.92	0.23
Baseline HBV pgRNA (<6.4 vs. ≥6.4 log copies/mL)	1.73	1.22–2.45	**0.002**	1.57	1.06–2.31	**0.023**
Liver cirrhosis (yes vs. no)	1.33	0.96–1.84	0.09	1.29	0.85–1.93	0.23

Significant *p* values are presented in bold. Abbreviations: ALT, alanine aminotransferase; CI, confidence interval; HBeAg, hepatitis B e antigen; HBsAg, hepatitis B surface antigen; HBV, hepatitis B virus; HR, hazard ratio; pgRNA, pregenomic RNA; ULN, upper limit of normal.

**Table 3 microorganisms-09-01146-t003:** Clinical characteristics of 20 chronic hepatitis B patients who achieved HBsAg loss after nucleos(t)ide analogue treatment and 141 chronic hepatitis B patients who received nucleos(t)ide analogue treatment without HBsAg loss for analyzing the factors associated with HBsAg loss.

Characteristics	CHB Patients Achieved HBsAg Loss after Nucleos(t)ide Analogue Treatment (*n* = 20)	CHB Patients Received Nucleos(t)ide Analogue Treatment without HBsAg Loss (*n* = 141)	*p* ^a^
Age at baseline (year) ^b^	48.73 ± 9.51	51.4 1± 11.97	0.34
Male (%)	17/20 (85%)	94/141 (66.7%)	0.12
HBeAg-positive at baseline ^b^	6/20 (30%)	45/141 (31.9%)	1.00
HBV genotype (B:C) ^c^	8:2	60:63	0.10
Baseline ALT (×ULN) ^b^	6.75 ± 11.03	3.43 ± 5.27	0.35
Baseline HBsAg (log IU/mL) ^b^	2.94 ± 1.28	3.17 ± 0.73	0.45
Baseline HBV DNA (log IU/mL) ^b^	5.99 ± 2.38	5.87 ± 1.68	0.88
Baseline HBV pgRNA (log copies/mL) ^b^	5.14 ± 2.05	5.08 ± 1.94	0.92
Liver cirrhosis (%)	2/20 (10%)	51/141 (36.2%)	**0.021**
Nucleos(t)ide analogue			
ETV (%)	18/20 (90%)	141/141 (100%)	
TDF (%)	2/20 (10%)	0 (0%)	
Treatment time (year)	5.12 (2.39–14.10)	4.91 (1.84–11.29)	0.24
Age at HBsAg loss (year)	54.45 ± 10.07		
HBV DNA at HBsAg loss (log IU/mL)			
not detected (%)	20/20 (100%)		
HBV pgRNA at HBsAg loss			
not detected (%)	0/20 (0%)		
<LOD (1466 copies/mL) (%)	7/20 (35%)		
≥LOD (1466 copies/mL) (%)	13/20 (65%) 5.90 (4.81–7.34) log copies/mL		
HBV pgRNA at the last follow-up (%)			
not detected (%)	0/20 (0%)		
<LOD (1466 copies/mL) (%)	20/20 (100%)		
Anti-HBs-positive at the last follow-up (%)	13/20 (65%)		

Continuous variables are expressed as mean ± standard deviation, except for treatment time and HBV pgRNA at HBsAg loss, which are expressed as median (range). Significant *p* values are presented in bold. Abbreviations: ALT, alanine aminotransferase; Anti-HBc, antibody to hepatitis B core antigen; Anti-HBs, antibody to hepatitis B surface antigen; CHB, chronic hepatitis B; ETV, entecavir; HBeAg, hepatitis B e antigen; HBsAg, hepatitis B surface antigen; HBV, hepatitis B virus; LOD, limit of detection; pgRNA, pregenomic RNA; TDF, tenofovir disoproxil fumarate; ULN, upper limit of normal. ^a^
*p* value compared between patients with HBsAg loss and patients without HBsAg loss. ^b^ Baseline means the time before receiving nucleos(t)ide analogue treatment. ^c^ HBV genotype was not determined in 10 patients with HBsAg loss and 18 patients without HBsAg loss because of low baseline HBV viral loads or limited baseline serum samples in these patients.

**Table 4 microorganisms-09-01146-t004:** Univariate and multivariate analysis of factors associated with HBsAg loss after nucleos(t)ide analogue treatment.

Factors	Univariate Analysis			Multivariate Analysis		
	HR	95% CI	*p* Value	HR	95% CI	*p*
Age (<50 vs. ≥50 years old)	1.23	0.51–2.97	0.64	9.81	1.23–77.94	**0.031**
Sex (male vs. female)	2.72	0.80–9.28	0.11	2.10	0.34–13.10	0.43
Baseline HBeAg (negative vs. positive)	1.19	0.46–3.10	0.72	1.53	0.11–20.81	0.75
HBV genotype (C vs. B)	0.25	0.05–1.17	0.08	0.41	0.06–3.02	0.38
Baseline ALT (× ULN)	1.04	0.99–1.10	0.13	1.00	0.92–1.10	0.97
Baseline HBsAg (<4 vs. ≥4 log IU/mL)	0.60	0.19–1.86	0.38	0.87	0.07–10.46	0.91
Baseline HBV DNA (<5 vs. ≥5 log IU/mL)	1.42	0.43–4.64	0.57	6.84	0.78–60.40	0.08
Baseline HBV pgRNA (<6.4 vs. ≥6.4 log copies/mL)	1.06	0.33–3.39	0.92	3.35	0.43–26.42	0.25
Liver cirrhosis (yes vs. no)	0.21	0.05–0.89	**0.035**	0.62	0.06–6.85	0.70
HBsAg decline within the first one year (≥1.5 vs. <1.5 log IU/mL)	17.57	6.99–44.17	**<0.001**	53.59	3.83–749.5	**0.003**

Significant *p* values are presented in bold. Abbreviations: ALT, alanine aminotransferase; CI, confidence interval; HBeAg, hepatitis B e antigen; HBsAg, hepatitis B surface antigen; HBV, hepatitis B virus; HR, hazard ratio; pgRNA, pregenomic RNA; ULN, upper limit of normal.

## Data Availability

The data that support the findings of this study are available from the corresponding author upon reasonable request.

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
