# Peer review of "Clinical Implications of Serum Hepatitis B Virus Pregenomic RNA Kinetics in Chronic Hepatitis B Patients Receiving Antiviral Treatment and Those Achieving HBsAg Loss"

_microorganisms, 2021, doi:10.3390/microorganisms9061146_

Round 1

Reviewer 1 Report

Major comments

No explanation is provided for the reason why in the low baseline HBV pgRNA group, its levels did not decrease after the initial increase and remained higher than  baseline. This has to be discussed.

Furthermore the fact that at HBsAg loss pgRNA remains detectable is of interest. How is this explained.  If cccDNA is still transcriptionally active why there is no production or excretion of HBsAg? With which method was HBsAg found undetectable?

It is surprising that such a number of HBeAg-negative patients remained serum pg-RNA positive under treatment. Is this genotype specific? Were these patients ever found to be HBeAg-positive? Was the specificity of the assay checked and if so how?

Minor comments

In materials and methods (section 2.1) is stated that “20 antibody to hepatitis B core antigen (anti-HBc)-positive individuals without past chronic HBV infection achieved HBsAg loss”. This is confusing. Needs to be rephrased ( 20 not treated antibody to hepatitis B core antigen (anti-HBc)-positive, HBsAg-negative  at presentation individuals). The whole paragraph has to be rewritten for clarity.

Figure 1 is illegible.

For studying the kinetics of serum HBV pgRNA, HBV DNA, and HBsAg during entecavir therapy, serum HBV pgRNA and HBV DNA quantification was performed at baseline, and the 3rd month (12th week), 6th month (24th week), 12th month (48th week), and 60th month (240th week) after treatment, and serum HBsAg quantification was per-formed at baseline, and the 1st year (48th week) and 5th year (240th week) after treatment. Virological response was defined as undetectable serum HBV DNA (< 60 IU/mL) after entecavir therapy.

In patients monitoring section (2.2), “timing of 60th month (240th week) after treatment” is 12 months post treatment or 60th month (240th week) of treatment? Please rephrase. Additionally, HBsAg quantification was performed  5 or 3 years post treatment? Again this paragraph has to be rewritten for clarity.

In section 3.1 paragraph 3 “HBsAg levels decreased gradually after entecavir therapy.” please add the word initiation: entecavir therapy initiation.

Section 3.5. “20 anti-HBc-positive individuals without past chronic HBV infection achieved HBsAg loss spontaneously” I think 20 only anti-HBc-positive individuals at presentation or 20 anti-HBc-positive, HBsAg-negative at presentation individuals would be more clear.

Reviewer 2 Report

Wu et al. present a manuscript concerning monitoring of the kinetics of hepatitis B virus pregenomic RNA levels in chronic hepatitis B patients receiving antiviral treatment. The purpose of the study was to check if detection of hepatitis B pregenomic RNA can be used as a marker for monitoring antiviral treatment in chronic hepatitis B patients.

The authors present work on an important topic . Identification of a new biomarker could be very helpful in the clinics. The manuscript is generally well written and the presented data is significant. However the presentation could benefit from some improvements.

Specific comments:

  1. Figure 1 is too small to read - should be taking one A4 page in a landscape orientation.
  2. Graphs in figure 2 are too small and most of the information in that figure is undecipherable. The font of the tables included in this figure has to be significantly increased, so it can be legible.
  3. The introduction is extremely short and could benefit from more information on already existing markers of infection and clinical outcome of chronic hepatitis B. I believe that the data presented by the authors are very interesting and contribute significantly to the field. In order to convey the significance of the addressed  question it would be important to describe in more detail the current state of the art with regards to pgRNA - how it is formed?; what are the known functions?; where it can be found?
  4. In the Materials and Methods part, in section 2.1 "Patient enrollment", the parts of the sentence refering to Part III and Part IV of the study are hard to understand - gramatically do not fit to the beginning of the sentence (second last sentence of the section 2.1).

Round 2

Reviewer 1 Report

The authors have answered all questions and made appropriate changes in this revised manuscript.

Minor comments

Since HBsAg loss was not confirmed by a qualitative (more sensitive than the quantitative) assay, this has to be discussed and included in the limitations of the study.
